# Characterization of the Lower Airways and Oral Microbiota in Healthy Young Persons in the Community

**DOI:** 10.3390/biomedicines11030841

**Published:** 2023-03-10

**Authors:** Fernando Sergio Leitao Filho, Carli Monica Peters, Andrew William Sheel, Julia Yang, Corey Nislow, Stephen Lam, Janice M. Leung, Don D. Sin

**Affiliations:** 1UBC Centre for Heart Lung Innovation, St. Paul’s Hospital, 1081 Burrard Street, Room 166, Vancouver, BC V6Z 1Y6, Canada; fernando.studart@hli.ubc.ca (F.S.L.F.); julia.yang@hli.ubc.ca (J.Y.); slam2@bccancer.bc.ca (S.L.); janice.leung@hli.ubc.ca (J.M.L.); 2Division of Respiratory Medicine, Department of Medicine, University of British Columbia, Vancouver, BC V1V 1V7, Canada; 3School of Kinesiology, Faculty of Education, University of British Columbia, Vancouver, BC V5Z 1M9, Canada; carli.peters@hli.ubc.ca (C.M.P.); bill.sheel@ubc.ca (A.W.S.); 4Faculty of Pharmaceutical Sciences, University of British Columbia, Vancouver, BC V1V 1V7, Canada; corey.nislow@ubc.ca; 5British Columbia Cancer Agency, Vancouver, BC V5Z 4E6, Canada

**Keywords:** 16S rRNA gene sequencing, healthy subjects, oral wash, bronchial brushing, bronchoalveolar lavage, microbiota, microbiome

## Abstract

Lower airway dysbiosis contributes to disease pathogenesis in respiratory diseases. However, little is known regarding the microbiota of lower airways or the oral cavity of healthy young persons. To address this gap, 25 healthy persons (24.3 ± 3.3 years; 52% females; no current smokers) underwent bronchoscopy during which bronchial brushing (BB) and bronchoalveolar lavage (BAL) fluid were collected. Prior to the procedure, an oral wash (OW) sample was also obtained. Microbiome analyses (16S rRNA locus) were performed (alpha- and beta-diversity, taxa annotations, and predicted functional metagenomic profiles) according to the airway compartment (BB, BAL, and OW). The greatest microbial richness was observed in OW and the lowest in BB (*p* < 0.001). Microbial communities differed significantly across compartments (*p* < 0.001), especially between BB and OW. Taxa analyses showed a significantly higher abundance of *Firmicutes* (BB: 32.7%; BAL: 31.4%) compared to OW (20.9%) (*p* < 0.001). Conversely, *Proteobacteria* predominated in OW (27.9%) as opposed to BB (7.0%) and BAL (12.5%) (*p* < 0.001), mostly due to a greater abundance of the bacteria in the *Haemophilus* genus in the OW (*p* < 0.001). The lower airway microbiota (BB and BAL) is significantly different from the OW microbiota in healthy young persons with respect to microbial diversity, taxa profiles, and predicted function.

## 1. Introduction

Currently, approximately 70% of bacterial species cannot be cultured using traditional culture-based methods [1]. These limitations have largely been overcome with the development of culture-independent methods, such as those that sequence the bacterial 16S ribosomal RNA (rRNA) locus [2]. Using these high-throughput methods, it has been shown that large and complex microbial ecosystems reside in different habitats along the human body, including the oral cavity and human airways. These ecosystems are known as the oral and lung microbiotas, respectively [3]. In health, there is a direct link between the oral and lung microbiota composition, as the oropharynx, through microaspiration, is the main source of the bacterial species commonly found in the airways [4,5]. 

A healthy resident microbiota in both the oral cavity and airway is important to prevent diseases. Possible mechanisms of this phenomenon include colonization resistance (in which the local microbiota prevents colonization of the mucosal surfaces by pathogens) [4], and regulation, in part, of the immune system [6]. Conversely, the loss of microbiota homeostasis and the subsequent development of microbial dysbiosis (i.e., a reduction in symbiotic bacteria in concert with a concomitant increase in pathogenic bacteria) [7] has been implicated in the pathogenesis of several diseases. Consistent with this notion, lung dysbiosis, reflected, for instance, by an increased abundance of *Pseudomonas* and *Moraxella* spp., has been linked with asthma [8], chronic obstructive pulmonary disease (COPD) [9], pulmonary fibrosis [10], and lung cancer [11]. Oral dysbiosis, on the other hand, which is mostly characterized by the appearance of *Porphyromonas gingivalis,* contributes to the development of dental caries, periodontal diseases, and oral cancer [12]. There is also a growing body of evidence supporting a link between oral and lung dysbiosis in different diseases [13,14]. Using a systematic and meta-analysis approach, Molina A et al. identified that periodontitis was significantly associated with a higher risk of COPD and obstructive sleep apnea and worse outcomes (need of assisted ventilation and mortality) following infection by SARS-CoV-2 [14]. Moreover, the local microbiota, depending on the microbial resident signatures, can also have prognostic implications. In keeping with this, we have shown previously that a higher relative abundance of *Staphylococcus* in the sputum of hospitalized patients for a COPD exacerbation was associated with a higher 1-year mortality risk [15]. Similarly, severe influenza infection was more common in children who demonstrated decreased microbial diversity and increased abundance, among others, of *Prevotella* species in the nasopharynx [16]. 

The majority of studies evaluating dysbiosis in lungs or oropharynx largely have recruited patients with specific diseases or those undergoing a clinical procedure (e.g., surgery). However, there is paucity of data on the “normal” oral and lung microbiota of healthy young persons [12,17]. Such information is essential for defining a microbial dysbiotic state within these habitats and serving as a crucial reference for comparative lung microbiome studies between “diseased” and healthy conditions. Here, we present the findings from the largest microbiome study of oral and lower airway microbiota in healthy young persons using oral wash and bronchoscopy samples. 

## 2. Materials & Methods

### 2.1. Subjects

This prospective study was conducted at the University of British Columbia (UBC, Vancouver, Canada). Eligible participants were male and female healthy volunteers, aged 18 years and older, with no previous diagnosis of any chronic medical conditions (except mild childhood asthma not requiring any maintenance inhalers) and normal lung function based on predicted values [18]. This study was approved by the Clinical Research Ethics Board at UBC (approval number: H14-00724), and all participants provided written informed consent. 

### 2.2. Procedures

Lower airway samples from subjects were obtained using bronchoscopy. Prior to the procedure, oral (OW) and bronchoscope channel (BCW) washes were collected. For the OW samples, participants were instructed to gargle with 10 mL of sterile 0.9% saline and then to spit into a sterile specimen cup. The BCW samples were retrieved by flushing 40 mL of sterile 0.9% saline through the bronchoscope channel into a similar specimen cup. After obtaining the OW and BCW samples, the participants were sedated, and topical lidocaine was administered to anesthetize the airways. Next, an Olympus^®^ bronchoscope was gently inserted, without any suctioning, into the oropharynx and subsequently into the left upper lobe. The bronchoscope was then wedged into a subsegmental bronchus (LB1 + 2 segment), after which a bronchial brush (BB) sample was collected from the 6–8th generation airways using a steel-tipped cytology brush (HOBBS Medical Inc.^®^, Stafford Springs, CT, USA). Upon collection, the BB sample was immediately immersed into an Eppendorf tube containing CytoLyt^®^ (Hologic^®^, Marlborough, MA, USA) for DNA preservation. A bronchoalveolar lavage (BAL) sample was then retrieved from the right upper lobe (RB3 segment). For this step, the first return was discarded (after instilling 20 mL of sterile 0.9% saline), and additional aliquots of 20–60 mL of sterile saline were instilled (up to a maximum of 200 mL) and then gradually suctioned up into a sterile cup (targeting 30 mL of lavage fluid).

### 2.3. Microbiome Profiling

PCR amplification was performed targeting the 16S rRNA gene V4 region using the Illumina MiSeq^®^ platform to generate 250-bp paired-end reads [19]. Additional details are provided in the online supplement. All samples were sequenced in a single sequencing run (batch) at the UBC Sequencing and Bioinformatics Consortium (Vancouver, Canada). Along with the clinical samples (BB, BAL, and OW), several control specimens were also included in our analyses to address any potential sources of contamination: extraction negatives (EN; contained only DNA extraction reagents), non-template controls (NTC; consisted of ultra-purified water used during the PCR reaction), CytoLyt controls (CC; contained only CytoLyt^®^ solution), and BCW samples (as described above).

The raw sequence data were analyzed using the QIIME 2™ (Quantitative Insights into Microbial Ecology) pipeline [20], version 2022.8, during which the Divisive Amplicon Denoising Algorithm (DADA2) [21] was applied to denoise the sequencing data, merge paired-end reads, and to cluster them into amplicon sequence variants (ASVs) [22]. To remove any potential non-bacterial DNA, the qiime quality-control exclude-seqs plugin function was applied to discard any sequences which did not align (percent identity of at least 80%) against the SILVA bacterial rRNA database (v138.1, Ref NR 99) [23]. Additionally, ASVs identified as potential contaminants by the Decontam R package [24] were also removed for downstream analyses. ASVs, which were not identified in at least two samples (across all samples, including controls) and whose taxa annotation (based on the SILVA database) was not available at the phylum level, were also discarded as those are likely due to sequencing errors. 

### 2.4. Microbiome Analyses

Oral (OW) and lower airway (BB and BAL) compartments were compared according to microbiome diversity and taxa analyses using customized in-house R scripts (version 4.1.1), based on functions provided by the Vegan (version 2.6.2) [25] and Phyloseq (version 1.38) [26] packages.

Alpha-diversity analyses (microbiome diversity within each sample) were performed using three different, yet complementary, metrics (microbial richness, the Shannon index, and the Pielou’s evenness). Briefly, richness relates to the number of different (unique) ASVs, whereas evenness reflects differences in the relative abundance of ASVs within each sample. Lastly, the Shannon index considers both microbial richness and evenness in its formula. Beta-diversity analyses (microbial diversity between samples) were also carried out using the Permutational Multivariate Analysis of Variance (PERMANOVA) test [27] using a generalized Unifrac distance matrix (alpha = 0.5) [28] in conjunction with principal coordinate analysis (PCoA) plots. All alpha- and beta-diversity analyses were performed in a rarefied feature table (13,834 reads across all clinical samples).

The relative abundance of the most common taxa at the phylum and genus levels according to specimen type was also determined. A Naive Bayes classifier trained on the SILVA rRNA database [23] was applied for taxonomic assignments. This was complemented by differential taxonomic analyses using LEfSe, which combines Kruskal–Wallis tests in association with a linear discriminant analysis (LDA) effect size to identify taxa differences between samples of interest [29]. LEfSe was also applied to compare the taxa annotations in bronchial brushings from our study (healthy participants) with similar samples from the British Columbia Cancer Agency (BCCA) cohort. Briefly, the BCCA cohort consisted primarily of older current or ex-smokers with or without COPD, who underwent bronchoscopy to investigate pulmonary nodules and masses (UBC approval ethics number: H14–03267) [11]. 

All sequencing data used in this study have been deposited to the National Center for Biotechnology Information’s Sequence Read Archive (SRA) under the BioProject number PRJNA918386.

### 2.5. Predicted Functional Metagenomics Analyses

PICRUSt2 (Phylogenetic Investigation of Communities by Reconstruction of Unobserved States, version 2.4.3) was applied to predict the functional metagenomics profiles based on the 16s rRNA gene sequencing data [30]. The PICRUSt2-inferred metagenome results were then expressed using the Kyoto Encyclopedia of Genes and Genomes (KEGG) pathways [31] and visualized and compared across airway compartments using LEfSe [29].

### 2.6. Statistical Analysis

Demographic data are expressed as mean ± standard deviation (SD) or numbers and percentages as appropriate. Conversely, alpha-diversity metrics and relative abundance data are expressed as median and interquartile ranges and were compared between microbiome compartments using a Kruskal–Wallis test. Adjusted *p*-values were determined using the Benjamini–Hochberg method [32]. The level of statistical significance was set at *p* < 0.05 (two-tailed) for all tests.

## 3. Results

### 3.1. Study Population

We prospectively evaluated 25 healthy subjects. The mean age of participants was 24.3 ± 3.3 years, and 13/25 were females (52%). Only one participant was a casual smoker, and two reported a previous history of mild asthma but were asymptomatic, had normal lung function, and did not use any medications at the time of bronchoscopy. Seven subjects (28%) were current or previous competitive swimmers. All participants showed normal spirometry values with regards to FVC (forced vital capacity), FEV_1_ (forced expiratory volume in one second), and FEV_1_/FVC ratio based on their predicted values [18]. Table 1 summarizes the baseline characteristics of the study population. All 25 participants underwent bronchoscopy without any complications. All bronchoscopy procedures were performed in a bronchoscopy unit located at the Vancouver General Hospital (Vancouver, BC, Canada).

### 3.2. 16S rRNA Gene Sequencing Data

The raw sequencing data contained a total of 12,116,921 reads across clinical and control samples. After denoising, chimera removal, ASV clustering, and filtering steps, 3,769,462 reads were retained for further analysis across 75 clinical samples (n = 25 for BB, BAL, and OW samples) (Figure 1 and Appendix A). The microbial structures (beta-diversity) of clinical samples were significantly distinct compared to those of controls (*p* < 0.001; Appendix A and Appendix A). For subsequent microbiome analyses, we only considered sequencing data related to the clinical samples (BB, BAL, and OW).

### 3.3. Oral and Lower Airway Microbiome Composition in Health

At the phylum level, in healthy young subjects, the majority of the reads in both oral and lower airway compartments belonged to *Bacteroidota*, *Firmicutes*, *Proteobacteria*, *Fusobacteriota*, and *Actinobacteriota*. These five phyla accounted for ~97%, ~96%, and ~98% of all reads across OW, BB, and BAL samples, respectively. At the genus level, the most abundant ones were *Prevotella-7*, followed by *Veillonella* and *Prevotella*. These three genera were responsible for ~50% of all the reads: 46% (OW), 50% (BB), and 48% (BAL). Figure 2, Figure 3 and Figure 4 show the mean relative abundances of the most common phyla and genera for each microbiome compartment using Krona plots [33] (pie charts in which different taxonomic levels are shown in sectors according to an embedded hierarchy). Overall, at the species level, the most abundant ASVs were identified as *Veillonella dispar*, *Haemophilus parainfluenzae*, followed by several *Prevotella* species (*P. jejuni*, P. *melaninogenica*, *P. pallens*, *P. salivae*, *P. intermedia*, and *P. nigrescens*) (Appendix A).

Interestingly, although our samples had been collected from healthy participants, ASVs corresponding to the typically pathogenic genera, such as *Moraxella*, *Pseudomonas*, *Staphylococcus,* and *Acinetobacter*, were also detected, albeit at a low relative abundance (Appendix A). These pathogenic genera were identified primarily in BB samples. For instance, the *Staphylococcus* and *Acinetobacter* genera were only observed in BBs, with each of these genera being detected in 5/25 of those samples (20.0%). 

### 3.4. Microbial Diversity Analyses

We detected several differences in the microbial diversity between BB, BAL, and OW samples. Overall, alpha-diversity analyses (Figure 5A–C) showed that the OW samples had the highest microbial richness, whereas the BB samples had the lowest (*p* < 0.001). Similar results, though in an opposing direction, were observed for evenness as the highest and lowest values were observed in the BB and OW samples, respectively (*p* < 0.001). For both microbial richness and evenness, BB and BAL samples differed significantly from each other (adj. *p* = 0.017 and adj. *p* = 0.034, respectively). No differences in the Shannon index were found across the sample types (*p* = 0.26).

Beta-diversity analyses revealed significant differences in the microbial composition between airway compartments (Figure 5D, *p* < 0.001). The lowest overlap in microbial communities was observed between the oral microbiota samples (OW) and the lower airway samples (BB and BAL; adj. *p*-value = 0.002 for both comparisons). Interestingly, the microbial communities were also significantly different between the BB and BAL compartments (adj. *p*-value = 0.004).

### 3.5. Differential Taxonomic Analyses

Several taxa-level differences were observed between microbiome compartments. At the phylum level (Table 2), both BB and BAL samples showed a significantly higher median relative abundance of *Firmicutes* (32.7% and 31.4%, respectively) compared to the OW samples (20.9%) (adj. *p* = 0.001 for both comparisons). Conversely, the *Proteobacteria* phylum predominated in the OW samples (median relative abundance: 27.9%) as opposed to the BB (7.0%) and BAL (12.5%) counterparts (adj. *p* < 0.001 for both comparisons). Among the top 15 genera (Table 3), the most striking difference was observed in the *Haemophilus* genus, which belongs to the *Proteobacteria* phylum. For this particular genus, the median relative abundance was 16.3% in the OW samples compared to 3.1% and 5.5% in the BB and BAL samples, respectively (adj. *p* < 0.001 for both comparisons). 

Taxa analyses according to LEfSe provided similar results (Figure 6A). This bioinformatics tool identified, from phylum to genus level, 36 differential taxa differential features across BB, BAL, and OW samples. Out of these, 19 taxa features predominated in BB, and most of them were Gram-positive bacteria. These included the *Actinobacteriota* phylum (including *Actinobacteria* class and *Micrococcales* order), the *Bacilli* class (along with *Staphylococcaceae* family and *Staphylococcus* genus), and the *Actinomycetales* order and its related *Actinomycetaceae* family and *Actinomyces* genus. On the other hand, seven taxa features were enriched in BAL, which included a mixture of Gram-positive (*Firmicutes* phylum) and Gram-negative (*Leptotrichiaceae* family and its corresponding *Leptotrichia* genus). Ten taxa features showed higher expression in OW, and all of these were Gram-negatives, including but not limited to the *Proteobacteria* phylum, *Gammaproteobacteria* class, and *Haemophilus* and *Neisseria* genera. 

### 3.6. Metagenome Prediction Analyses

PICRUSt2 analyses were applied to our 16S amplicon sequencing data to predict the metagenome functions of the bacterial microbiota in oral (OW) and lower airway (BB and BAL) environments. Using the PICRUSt2 output in association with LEfSe (LDA score > 2), we identified 58 pathways, which showed significantly different expressions across these compartments: BB (n = 19); BAL (n = 13); and OW (n = 26) (Figure 6B). In addition to differences in metabolic pathways (e.g., those related to metabolism and degradation of carbohydrates, amino acids, and lipids), some pathway differences between these samples are noteworthy from a clinical standpoint. For instance, pathways involved with cell motility (bacterial chemotaxis and flagellar assembly), biofilm formation (*Vibrio cholerae* pathogenic cycle), and biosynthesis of specific metabolites (derived from penicillin and cephalosporin) were enriched in BB. On the other hand, pathways related to membrane transport (bacterial secretion system), metabolism of cofactors and vitamins (lipoic acid metabolism), and glycan biosynthesis and metabolism (lipopolysaccharide biosynthesis) were more highly expressed in OW. 

### 3.7. Lower Airway Microbiota Changes Associated with Local Dysbiosis

The taxa composition in bronchial brushings of healthy young subjects was compared against similar samples from the BCCA cohort (Appendix A), which consisted of healthy older smokers and patients with COPD. For better plot visualization, LEfSe analyses were performed using a strict LDA score (>4.0). Notwithstanding, 25 differential taxa features were still identified between healthy subjects and participants from the BCCA cohort (Figure 7). Out of these, 13 taxa were enriched in the former group, including, among others, the *Prevotella*, *Leptotrichia*, *Neisseria*, and *Haemophilus* genera. Only two taxa predominated in healthy older smokers (*Prevotellaceae* UCG-001 genus and *Comamonadaceae* family). Lastly, 10 taxa showed higher expression in COPD, including the *Pseudomonas* and *Streptococcus* genera, which represent markers of airway dysbiosis and belong to potentially pathogenic bacteria (PPB) involved with lung infections.

## 4. Discussion

Here, we describe in detail the oral and lower airway microbiota observed in young healthy persons (24.3 ± 3.3 years). According to our data, the most abundant phyla, expressed in median relative abundance, were *Bacteroidota* (OW: 41.7%, BB: 42.3%; BAL: 40.2%), *Firmicutes* (OW: 20.9%; BB: 32.7%; BAL: 31.4%), and *Proteobacteria* (OW: 27.9%; BB: 7.0%; BAL: 12.5%). Overall, at the genus level, *Prevotella* (considering both *Prevotella* and *Prevotella-7* genera) showed a median relative abundance of 30.7% (OW), 35.0% (BB), and 30.3% (BAL), followed by *Veillonella*, with 15.3% (OW), 15.5% (BB), and 16.8% (BAL). Our results are in line with previous studies. Dickson et al. observed a high proportion of *Veillonella* and *Prevotella* species in oral rinse specimens collected from eight healthy subjects (mean age: 53 ±15 years) [34]. Ramsheh et al. investigated the lower airway microbiota using bronchial brushings from 207 healthy older subjects (52 to 66 years) [35]. These authors reported a higher expression of both the *Bacteroidetes* phylum (also known as *Bacteroidota*) and the *Prevotella* genus, as these taxa showed a median relative abundance of 54.0% and 47.7%, respectively. Whether age may account, at least partially, for these differences in relative abundances between the former study and ours is unknown. We have extended the findings from these previous studies by providing information regarding the taxa profiles at the species level. Overall, the most abundant ASV in our samples belonged to the *Veillonela* genus (*V. dispar*), and this species has been reported to colonize the tongue of healthy individuals [36]. Concerning the *Prevotella* species, *P. melaninogenica*, *P. nigrescens*, and *P. pallens* are ubiquitous species found in the oral cavity from birth [37]. *P. jejuni*, despite its association with celiac disease, is also considered an oral resident species [37]. All these *Prevotella* species were not only detected in our study but were also the most abundant ones related to the *Prevotella* genus. It is worth mentioning that *Veillonella* spp. are Gram-negative diplococci, while *Prevotella* spp. are anaerobic Gram-negative rods. Both of these organisms are colonizers of the normal airways, and their presence in the lower airway microbiota is due to microaspiration from the oropharynx [4,34]. For the *Haemophilus* genus, *Haemophilus parainfluenza* was the most frequent species detected across our samples; this species can be isolated from the upper respiratory tract microbiota of healthy subjects, as well as in the microbiota of patients with sinusitis and acute exacerbations of COPD [38]. 

Among our healthy participants, ASVs belonging to the *Moraxella*, *Pseudomonas*, *Staphylococcus*, and *Acinetobacter* genera were also identified. Several members of these genera are associated with serious human diseases, including pneumonia and endocarditis, thus their presence in lung samples of healthy participants should be interpreted cautiously. One possible explanation may be contamination of our bronchoscopy specimens with bacterial DNA present in laboratory reagents. However, none of these sequences were flagged by the Decontam contamination removal process [24], supporting the notion that these sequences represent real biological data rather than cross-contamination. It is intriguing that these pathogenic genera were mostly isolated in bronchial brushings. Whether the normal airway microbiome might harbor pathogenic genera in very low abundances from intermittent colonization is plausible. In keeping with this, a previous study also detected these genera at very low abundances in bronchoscopy samples from healthy subjects [35]. Conversely, in COPD, chronic inflammation and disruption of the mucociliary clearance typically lead to lower airway colonization by PPB (i.e., lung microbial dysbiosis) [5], such as *Streptococcus pneumoniae*, *Staphylococcus aureus*, and *Pseudomonas aeruginosa*. Interestingly, smoking does not appear to significantly shift the lower airway microbiome unless it is associated with early COPD [5]. These findings are consistent with our results (Figure 7). 

Our data also revealed significant differences across oral and lower airway microbiome compartments, and the most pronounced differences in microbial diversity were detected between the oral washes and bronchial brushings. These differences between the oral and lower airway microbiota are partially explained by the amount of biomass in each of these microenvironments; the oral microbiota represents a high-biomass milieu compared to the relatively low biomass found in the lower airways. Consistent with this, the median microbial richness in OW (150) was 54% and 14% higher compared to that observed in bronchial brushings (94) and BAL (131). Interestingly, the microbial communities were also significantly distinct between related lower airway compartments (bronchial brushings and BAL fluid). These differences in microbial diversity between BB and BAL samples may be explained by two factors: (1) whereas bronchial brushing evaluates approximately 1 cm^2^ of the airway mucosa, a BAL fluid aliquot is capable of sampling a much larger airway area [34]; and (2) the BB samples were collected from LB1 + 2 as opposed to RB4 or RB5 for the BAL. As previously shown by Erb-Downward et al. [39], it is possible that the bacterial microbiota may be influenced by regional differences in lungs within the same individual (e.g., sample collection from different lung segments).

A few studies have investigated interventions that can be applied to reverse at least in part the local dysbiosis in the oral cavity and lower airways of the diseased lung. Given the ease of access, most of those studies have addressed interventions on pathogens, such as those related to the orange- and red-complexes, which are associated with oral dysbiosis and periodontal disease [40]. For instance, Butera et al. have shown a bacterial load decrease of orange-complex pathogens (*Fusobacterium*, *Prevotella* and *Campylobacter* spp.) after a 6-month treatment with a probiotics-based toothpaste in association with probiotics-based chewing gum containing bacteria belonging to the *Lactobacillus* and *Bifidobacterium* genera [41]. In terms of potential therapeutic interventions targeting airway dysbiosis, reducing airway concentration levels of indole-3-acetic acid (IAA), derived from *Lactobacillus* spp., upregulated interleukin-22 signaling and epithelial cell apoptosis pathways in COPD [42]. In the same study, the authors observed an increase in local IAA levels with a subsequent restoration of their airway protective effects in mice following intranasal inoculation of two airway lactobacilli.

Our study has several limitations. First, no a priori sample size calculation or power analysis was performed owing to insufficient prior data on this topic involving young healthy volunteers. The sample size was empirically derived over a 2-year period based on the number of young, healthy adults who were willing to participate in a bronchoscopy study in Vancouver (and terminated just prior to the start of the COVID-19 pandemic). While the study’s sample size is relatively low (n = 25), it is still one of the largest studies of its kind. Second, our research subjects were young adults, thus these data might not apply to older persons. However, as co-morbidities and harmful environmental exposures (e.g., cigarette smoke and air pollution) increase with the ageing process, ascertaining data in “healthy” older persons may be challenging. Third, as previously described, we sampled only two areas in the lower respiratory tract. As the lower airways have a complex branching pattern, it is possible that there could be significant sampling error in our measurement. 

## 5. Conclusions

Our study describes in detail the taxa profiles and microbial diversity along with predicted metagenomic functions in both the oropharynx and lower airways of healthy young individuals providing an important reference dataset for these habitats, against which future analyses can be benchmarked. Our data support the notion that the lower airway microbiota differs significantly from the oral microbiota in healthy young persons and that significant differences may also be observed between bronchial brushings and BAL fluid even when collected from the same subject. By better understanding the expected taxa profiles in health, it will be possible to properly characterize homeostatic host–microbe interactions and perturbations in different habitats. New studies will be needed to investigate how these host–microbe interactions contribute to diseases in dysbiotic environments and how to revert dysbiosis using medical interventions. 

## Figures and Tables

**Figure 1 biomedicines-11-00841-f001:**
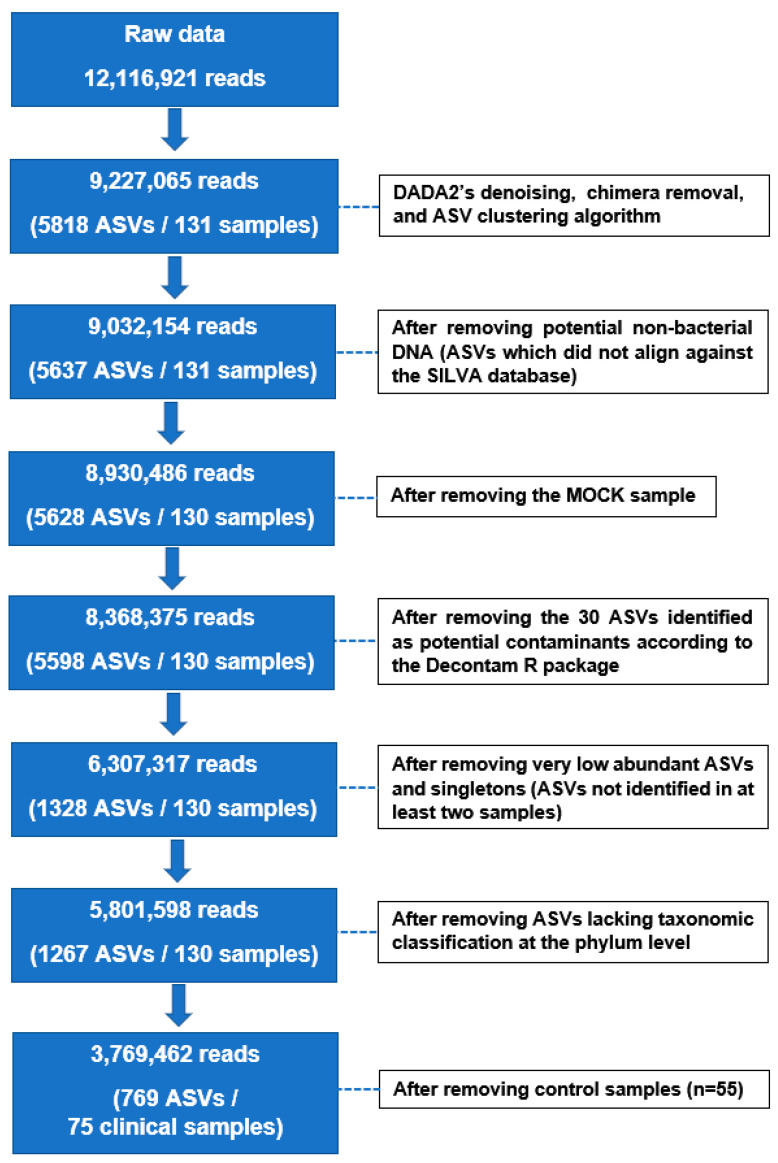
Flowchart showing the number of reads and amplicon sequence variants retained across clinical samples for microbiome analyses after denoising, filtering steps, and removal of controls. Control samples included extraction negatives (contained only DNA extraction reagents), non-template controls (consisted of ultra-purified water used during the PCR reaction), CytoLyt controls (contained only CytoLyt^®^ solution), and bronchoscope channel wash samples (retrieved by flushing 40 mL of sterile 0.9% saline through the bronchoscope into a specimen cup before bronchoscopy). Definition of abbreviations: ASV = amplicon sequence variant.

**Figure 2 biomedicines-11-00841-f002:**
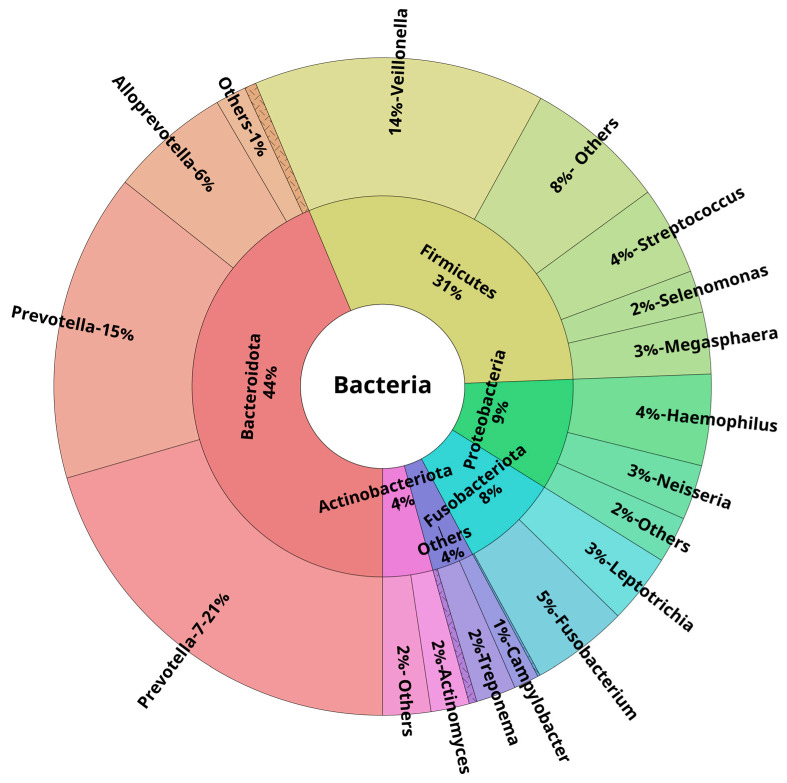
Krona plot showing the mean taxa relative abundances at the phylum and genus levels in bronchial brushings of healthy young subjects (n = 25). Only the phylum (inner layer) and genus (outer layer) levels are displayed for better visualization. Only the top 15 genera are shown (based on their mean relative abundance across all clinical samples, n = 75). Less abundant phyla and genera were included in the “Others” group.

**Figure 3 biomedicines-11-00841-f003:**
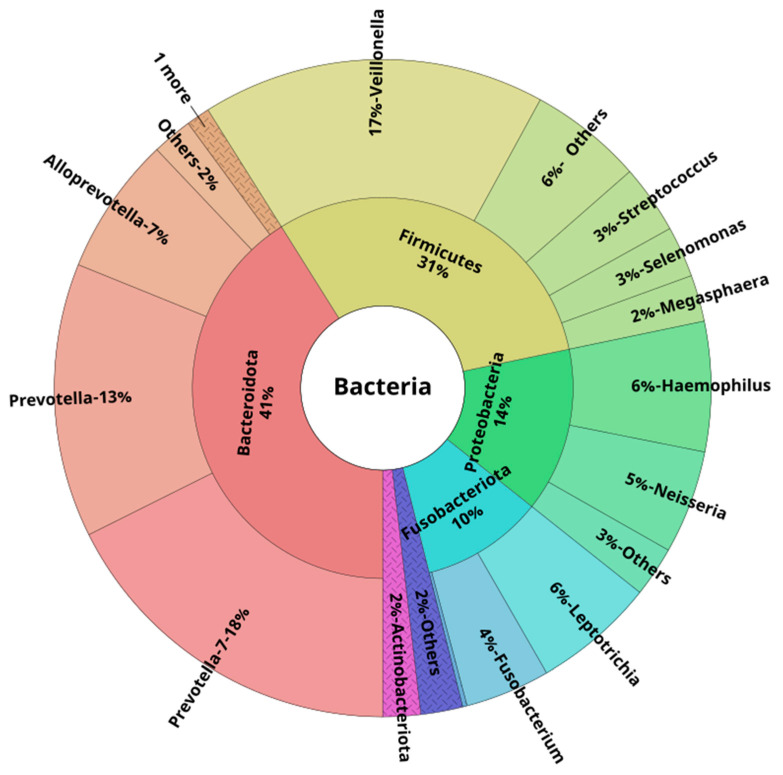
Krona plot showing the mean taxa relative abundances at the phylum and genus levels in bronchoalveolar lavage samples of healthy young subjects (n = 25). Only the phylum (inner layer) and genus (outer layer) levels are displayed for better visualization. Only the top 15 genera are shown (based on their mean relative abundance across all clinical samples, n = 75). Less abundant phyla and genera were included in the “Others” group.

**Figure 4 biomedicines-11-00841-f004:**
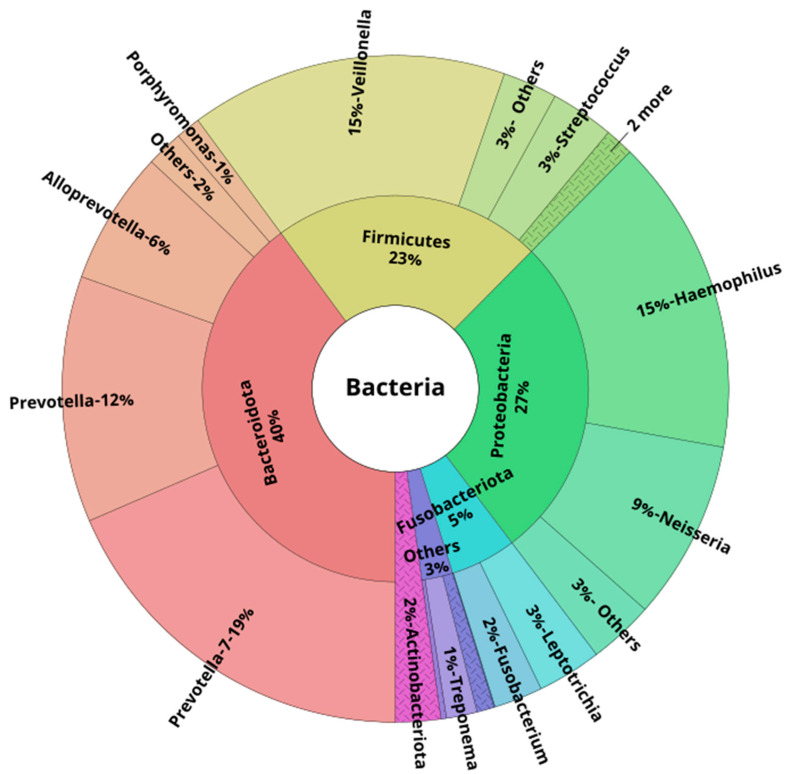
Krona plot showing the mean taxa relative abundances at the phylum and genus levels in oral washes of healthy young subjects (n = 25). Only the phylum (inner layer) and genus (outer layer) levels are displayed for better visualization. Only the top 15 genera are shown (based on their mean relative abundance across all clinical samples, n = 75). Less abundant phyla and genera were included in the “Others” group.

**Figure 5 biomedicines-11-00841-f005:**
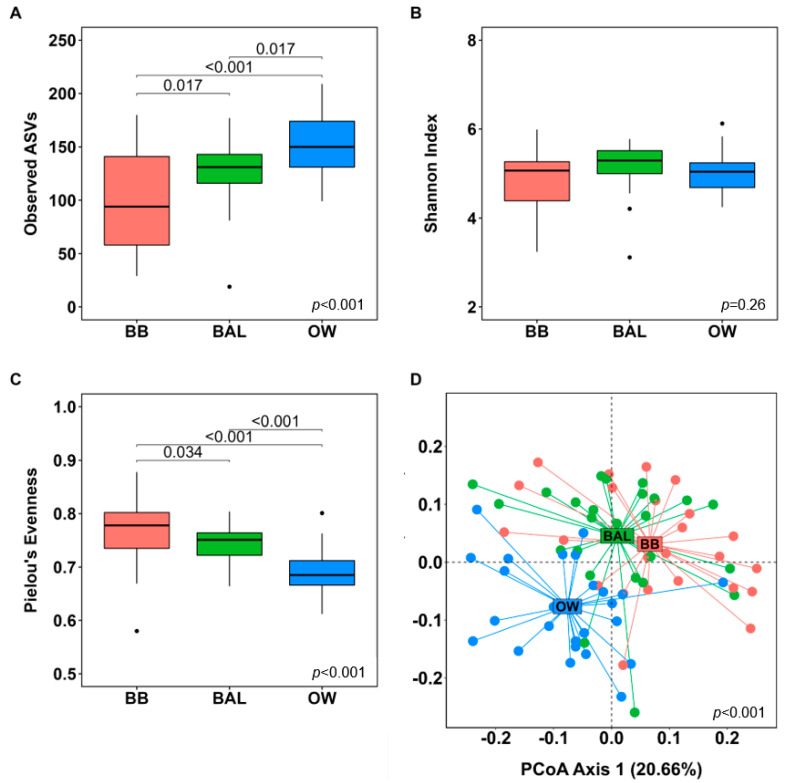
Alpha- and beta-diversity analyses across oral and lower airway compartments of healthy young subjects (n = 25). Panels (**A**–**C**): Alpha-diversity analyses using three different metrics between bronchial brushings (BB), bronchoalveolar lavage (BAL), and oral wash (OW) samples (n = 25 for each microbiome compartment). *p*-values were calculated using a Kruskal–Wallis test; adjusted *p*-values were obtained according to the Benjamini–Hochberg method. Panel (**D**): Principal Coordinates Analysis (PCoA) plot showing a comparison of microbial structures (beta-diversity) across microbiome compartments based on a generalized Unifrac distance matrix. *p*-value was calculated using the Permutational analysis of variance (PERMANOVA) method based on the *adonis* function of the vegan R package; pairwise PERMANOVA results (according to the Benjamini–Hochberg procedure): BB vs. BAL: adj. *p*-value = 0.004; BB vs. OW: adj. *p*-value = 0.002; BAL vs. OW: adj. *p*-value = 0.002. Definition of abbreviations: ASV = amplicon sequence variant.

**Figure 6 biomedicines-11-00841-f006:**
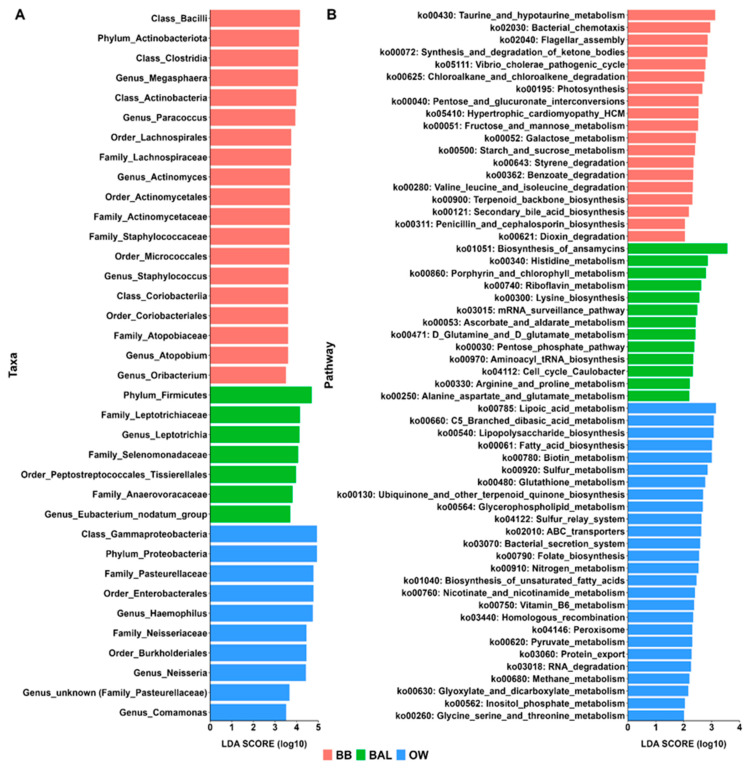
Differential taxa and metagenomic analyses across oral and lower airway compartments of healthy young subjects (n = 25). Panel (**A**): LEfSe analyses, based on a linear discriminant analysis (LDA) effect size > 3, identified several differential taxa features across microbiome compartments. LEfSe analyses were carried out considering five taxonomic levels (from phylum to genus level); Panel (**B**): Comparison of functional pathway predictions (based on the Kyoto Encyclopedia of Genes and Genomes database) by PICRUSt2; differential functional pathways were visualized between microbiome compartments using LEfSe with an LDA effect size > 2.

**Figure 7 biomedicines-11-00841-f007:**
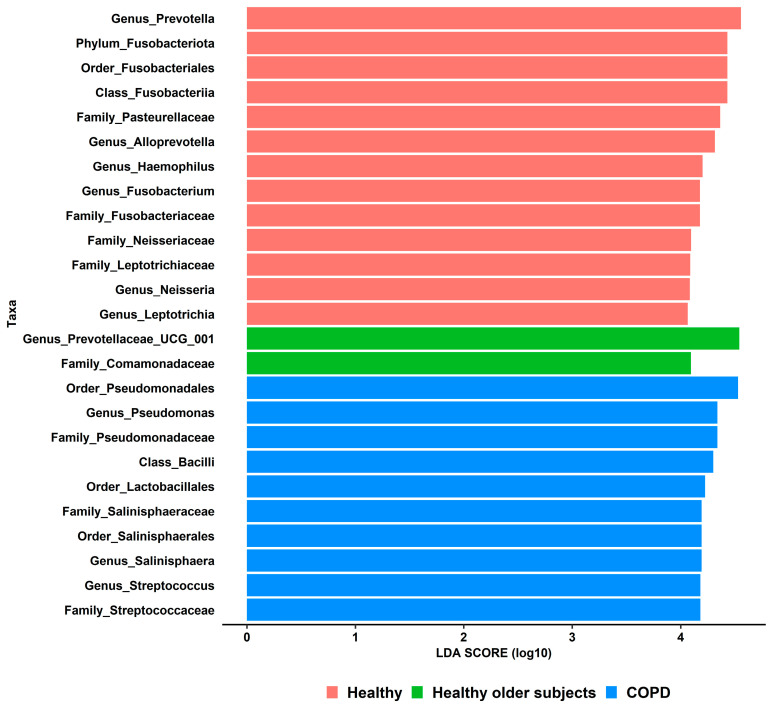
Differential taxa features identified in bronchial brushings of healthy young (n = 25), healthy older (n = 23), and COPD (n = 24) subjects. LEfSe analyses were performed using a linear discriminant analysis (LDA) effect size > 4.0. Both COPD (chronic obstructive pulmonary disease) and healthy older subjects were enrolled into the British Columbia Cancer Agency (BCCA) cohort.

**Table 1 biomedicines-11-00841-t001:** Baseline characteristics of healthy young participants.

Variables	Participants (n = 25)
Age, years	24.3 ± 3.3
Sex, female	13 (52)
Current/previous smoker	1 (4)
Swimmers	7 (28)
BMI, kg/m^2^	22.57 ± 1.96
FVC, liters	5.4 ± 0.94
FVC, % of predicted	111.43 ± 11.86
FEV_1_, liters	4.39 ± 0.77
FEV_1_, % of predicted	106.09 ± 11.38
FEV_1_/FVC, %	81.47 ± 6.53

Values are expressed as mean ± SD or n (%) unless otherwise noted. Definition of abbreviations: BMI = body mass index; FVC = forced vital capacity; FEV_1_ = forced expiratory volume in one second.

**Table 2 biomedicines-11-00841-t002:** Relative abundance comparisons at the phylum level across oral and lower airway compartments of healthy young subjects (n = 25).

Taxon	MICROBIOME COMPARTMENT						*p*-Value *	
	Bronchial Brushings		BAL Samples		Oral Wash Samples			
Phylum	Median	Q1; Q3	Median	Q1; Q3	Median	Q1; Q3	*p*	adj. *p*
*Bacteroidota*	42.3	36.8; 46.2	40.2	37.3; 45.5	41.7	35.2; 44.5	0.59	0.59
*Firmicutes*	32.7	22.8; 37.6	31.4	26.9; 35.9	20.9	17.5; 25.5	<0.001	<0.001
*Proteobacteria*	7.0	4.6; 14.7	12.5	7.7; 18.5	27.9	23.0; 32.8	<0.001	<0.001
*Fusobacteriota*	6.2	3.0; 10.3	9.1	6.4; 14.0	4.6	3.1; 7.1	0.011	0.014
*Actinobacteriota*	3.7	2.9; 5.1	2.1	1.4; 2.4	1.7	1.0; 2.5	<0.001	<0.001

Values expressed in %. All phyla whose relative abundances were lower than 2% across all clinical samples are not shown (based on their mean relative abundance across all clinical samples). Definition of abbreviations: BAL = bronchoalveolar lavage; Q1 = first quartile; Q3 = third quartile. * *p*-values were calculated using a Kruskal–Wallis test; adjusted *p*-values obtained according to the Benjamini–Hochberg method.

**Table 3 biomedicines-11-00841-t003:** Relative abundance comparisons of the most common genera across oral and lower airway compartments of healthy young subjects (n = 25).

Taxon	MICROBIOME COMPARTMENT						*p*-Value *	
	Bronchial Brushings		BAL Samples		Oral Wash Samples			
Genus	Median	Q1; Q3	Median	Q1; Q3	Median	Q1; Q3	*p*	adj. *p*
*Prevotella-7*	21.0	16.2;25.0	18.0	15.5; 20.7	19.7	15.7; 21.8	0.29	0.40
*Veillonella*	15.5	9.4; 17.9	16.8	13.9; 21.2	15.3	11.4; 16.8	0.28	0.40
*Prevotella*	14.0	9.4; 16.0	12.3	9.8; 14.1	11.0	8.8; 13.9	0.39	0.48
*Haemophilus*	3.1	1.8; 7.2	5.5	4.7; 7.2	16.3	12.2; 18.7	<0.001	<0.001
*Alloprevotella*	4.8	2.3; 10.0	7.2	4.4; 8.6	5.2	3.8; 10.1	0.47	0.51
*Neisseria*	1.7	0.5; 5.0	4.2	1.5; 7.1	7.1	3.7; 11.5	<0.001	0.001
*Leptotrichia*	1.8	1.0; 4.5	5.7	2.7; 7.7	2.6	1.4; 4.0	0.006	0.014
*Fusobacterium*	3.1	1.8; 6.2	2.5	1.9; 3.9	1.5	1.0; 2.9	0.07	0.12
*Streptococcus*	4.3	2.6; 5.7	3.2	1.9; 4.1	2.2	1.7; 4.1	0.06	0.11
*Megasphaera*	2.7	0.3; 4.2	2.4	1.5; 3.2	0.5	0.2; 0.8	<0.001	0.002
*Selenomonas*	1.2	0.0; 2.4	2.1	1.0; 3.7	0.7	0.4; 1.1	0.004	0.012
*Treponema*	0.7	0.3; 1.0	0.6	0.2; 0.9	0.3	0.1; 1.2	0.48	0.51
*Actinomyces*	1.8	1.2; 2.5	1.0	0.7; 1.3	0.9	0.7; 1.3	0.003	0.01
*Porphyromonas*	0.4	0.1; 0.9	0.9	0.2; 1.6	0.9	0.4; 1.7	0.03	0.08
*Campylobacter*	0.8	0.5; 1.2	0.9	0.5; 1.3	0.7	0.6; 1.1	0.75	0.75

Values expressed in %. Only the top 15 genera are shown. Definition of abbreviations: BAL = bronchoalveolar lavage; Q1 = first quartile; Q3 = third quartile. * *p*-values were calculated using a Kruskal–Wallis test; adjusted *p*-values obtained according to the Benjamini–Hochberg method.

## Data Availability

All sequencing data used in this study have been deposited to the National Center for Biotechnology Information’s Sequence Read Archive (SRA) under the BioProject number PRJNA918386. Additional data supporting the findings of this study are available from the corresponding author upon reasonable request.

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
