# Peer review of "Characterization of the Lower Airways and Oral Microbiota in Healthy Young Persons in the Community"

_biomedicines, 2023, doi:10.3390/biomedicines11030841_

Round 1

Reviewer 1 Report

The manuscript by Fernando Sergio Leitao Filho et al. describes the microbiota of the lower and oral airways in young community patients.

This manuscript lacks novelty, and suffers from defects that should be corrected before a possible publication.

Global: italicize "et al.", "ie.", bacterial names

Prefer passive forms.

Introduction: see to discuss the impact of acute situations on the microbiome ( for example, influenza -S.Langevin et al. JGV 2017-, Metaanalysis - D.T.J. Broderick Frontiers Microbiol 2021)

Methods: Including women of childbearing age induces a bias related to the menstrual cycle that should be explored/detailed (time of cycle etc.). 

How was the number of subjects to be included determined a priori (the number of subjects is rather low). 

The patients with asthma in the enance having been treated (even temporarily) had an impacted microbiome, the study must take this into account and exclude them. 

What was the procedure for recruiting patients, where did they come from etc....

Bioinformatics : The last sentence of the "microbiome profiling" part should be justified, as these rare species may be of interest.

Taxonomic analysis. The phyla have been renamed, please take this into account when writing the article.

Figure 2: A krona will be more readable/interpretable.

Author Response

RESPONSE TO REVIEWERS’ COMMENTS

REVIEWER: 1

Q1. Global: italicize "et al.", "ie.", bacterial names.

R1. We have performed a thorough revision and changed to italics any parts containing "et al.", "i.e.", and bacterial names accordingly.

Q2. Prefer passive forms.

R2. Whenever applicable, we have changed from the active to the passive form as suggested by the Reviewer.

Q3. Introduction: see to discuss the impact of acute situations on the microbiome (for example, influenza -S.Langevin et al. JGV 2017-, Metaanalysis - D.T.J. Broderick Frontiers Microbiol 2021).

R3. As suggested by the Reviewer, we have extended the Introduction and included information on how the local microbiota might contribute or respond to acute situations (Langevin S et al. [16] – page 3, lines 4-6).

Q4. Methods: Including women of childbearing age induces a bias related to the menstrual cycle that should be explored/detailed (time of cycle etc.).

R4. We agree with the Reviewer that including childbearing age represents a confounding factor, as sex hormones might interfere with microbiota in specific habitats. Hormone fluctuations due to the menstrual cycle phase are an inherent aspect of healthy young female biology, and excluding females would fail to account for the biological differences between females and males. In our study, female participants were tested randomly throughout their menstrual cycle, and oral contraceptives were not an exclusion criterion.

Thus, excluding women from participation in our study due to this concern would add more bias to our results/interpretation than possible benefits. First, to the best of our knowledge, it is unclear whether sex hormones play an important role in the oral and airway microbiota composition. In keeping with this, in a previous study from our group, after removing the ovaries, we observed that the cecal microbiome of ovariectomized female mice resembled those of male mice (Tam A et al. PLoS One. 2020 Apr 6;15(4):e0230932.), although this effect was not observed on the lung microbiome (homogenized lung tissue) - unpublished data. Second, the main purpose of our study is to describe the oral and airway microbiome in healthy young persons, which include both males and females. Including only healthy male volunteers would add a selection bias and further limit the purpose of our study (to provide an important reference dataset for both the lower airways and mouth microbiomes against which future analyses can be benchmarked). Third, 13/25 subjects (52%) included in our study were females, and their exclusion would significantly decrease our sample size, which is also a concern of the Reviewer, as highlighted in the next question (Q5). Fourth, we performed additional microbiome analyses, and for all three specimen types we evaluated (oral washes, bronchial brushings, and BAL), the microbial communities between female and male participants did not differ significantly from each other (the PCoA plots are included in the attached word file).

Q5. How was the number of subjects to be included determined a priori (the number of subjects is rather low).

R5. We thank the Reviewer for bringing this up. First, our sample size was limited to the challenges of recruiting healthy young subjects to undergo invasive procedures (including a research bronchoscopy) in our study. To put this into perspective, we needed approximately two years to recruit 25 volunteers. Second, the main purpose of this study is to describe the oral and lower airway microbiota in healthy young persons (age: 19-30 years), as this data is scarce in the literature. For instance, both relevant papers suggested by this Reviewer evaluated specifically pediatric participants (age < 18 years). We should also mention that, given the descriptive nature of our study, a sample size calculation and power analysis became even more complex. Moreover, the best approach to estimate a sample size calculation when performing microbiome analysis remains unclear. We have added a sentence in the Discussion section emphasizing this limitation (page 15, lines: 1-5).

Q6. The patients with asthma in the enance having been treated (even temporarily) had an impacted microbiome, the study must take this into account and exclude them.

R6. Only two participants reported a history of very mild asthma, requiring no medications including rescue inhalers. Thus, we do not think there is a sufficient clinical or biological reason for the exclusion of these two subjects. Additionally, their exclusion would further decrease our sample size.

Q7. What was the procedure for recruiting patients, where did they come from etc....

R7. These volunteers were recruited at the School of Kinesiology, University of British Columbia, Vancouver, Canada, as they had participated voluntarily in previous studies at the same research centre.

Q8. Bioinformatics : The last sentence of the "microbiome profiling" part should be justified, as these rare species may be of interest.

R8. The last sequence of the Microbiome profiling section has been changed to: “ASVs, which were not identified in at least two samples (across all samples, including controls) and whose taxa annotation (based on the SILVA database) was not available at the phylum level, were also discarded, as those are likely due to sequencing errors.The underlined part has been added to the revised manuscript, justifying the exclusion of those specific ASVs. 

Q9. Taxonomic analysis. The phyla have been renamed, please take this into account when writing the article.

R9. In our manuscript, all taxa annotations were retrieved directly from the SILVA ribosomal RNA gene database. To the best of our knowledge, the last version is 138.1 (release data: 02-Nov-2020),  which was applied in our analyses (https://www.arb-silva.de/download/archive/).

Q10. Figure 2: A krona will be more readable/interpretable.

R10. The Results section, including its tables and figures, has been completely rearranged, making it easier for the reader to understand the data presented in our manuscript. As recommended by the Reviewer, we have replaced bar plots with krona plots to describe both the oral and lower airway microbiota composition in our study.

Reviewer 2 Report

Manuscript of considerable interest for the dental sector which requires a minor revision

Well described abstract

Few keywords, add specific ones

Introduction: how does the oral microbiota change? based on respiratory needs

Materials and methods; How was the sample size calculated?

Very confusing results: rearrange them to make it easier for the reader

Discussion; add as future objectives, the use of probiotics, paraprobiotics and postbiotics to maintain a state of homeostasis of the oral cavity, as already studied by the research group of Prof Scribante

Conclusions; add proactive action

Bibliography: add references required

Author Response

RESPONSE TO REVIEWERS’ COMMENTS

REVIEWER: 2

Manuscript of considerable interest for the dental sector which requires a minor revision.

Q1. Well described abstract.

R1. We thank the Reviewer for these kind words.

Q2. Few keywords, add specific ones.

R2. We have changed the keyword count from four to seven related to our abstract and also chosen more specific ones. Now we are using the following keywords: 16S rRNA gene sequencing; healthy subjects; oral wash; bronchial brushing; bronchoalveolar lavage; microbiota; microbiome.

Q3. Introduction: how does the oral microbiota change? based on respiratory needs.

R3. The Introduction has been significantly expanded, especially the second paragraph, in which we discuss microbial changes associated with oral dysbiosis, also providing evidence linking oral and lung dysbiosis in specific respiratory diseases (page 2, lines 10-24; page 3, lines: 1-6).

Q4. Materials and methods; How was the sample size calculated?

R4. We thank the Reviewer for bringing this up. First, our sample size was limited to the challenges of recruiting healthy young subjects to undergo invasive procedures (including a research bronchoscopy) in our study. To put this into perspective, we needed approximately two years to recruit 25 volunteers. Second, the main purpose of this study is to describe the oral and lower airway microbiota in healthy young persons (age: 19-30 years), as this data is scarce in the literature. We should also mention that, given the descriptive nature of our study, a sample size calculation and power analysis became even more complex. Moreover, the best approach to estimate a sample size calculation when performing microbiome analysis remains unclear. We have added a sentence in the Discussion section emphasizing this limitation (page 15, lines: 1-5).

Q5. Very confusing results: rearrange them to make it easier for the reader.

R5. The Results section, including its tables and figures, has been completely rearranged, making it easier for the reader to understand the data presented in our manuscript.

Q6. Discussion; add as future objectives, the use of probiotics, paraprobiotics and postbiotics to maintain a state of homeostasis of the oral cavity, as already studied by the research group of Prof Scribante.

R6. A new paragraph has been added to the Discussion (page 14, lines 11-23), highlighting possible interventions against oral and lower airway dysbiosis. Among the references included in this paragraph, there is one study discussing specifically the effects of probiotics in periodontitis and oral dysbiosis (Butera et al. [41] – page 14, lines 14-18).  

Q7. Conclusions; add proactive action.

R7. An additional sentence has been added to the Conclusion, supporting a proactive action toward new microbiome studies to better characterize host-microbe interactions in health and diseases and also investigate medical interventions to revert dysbiosis (page 16, lines 18-22). 

Q8. Bibliography: add references required.

R8. Additional references, especially in regard to the impact of probiotics in oral dysbiosis, have been added to the Bibliography accordingly.

Reviewer 3 Report

The study is an original and interesting one, however, several issues must be addressed. Please see the enclosed PDF.

Author Response

RESPONSE TO REVIEWERS’ COMMENTS

REVIEWER 3

The study is an original and interesting one, however, several issues must be addressed. Please see the enclosed PDF.

Q1. The introduction is too brief. The authors should expand on the subject at hand.

R1. The Introduction has been significantly expanded, especially the second paragraph, as recommended by the Reviewer (please see R2).

Q2. The authors should also discuss the link between COVID-19 and periodontitis and how a poor oral status could lead to a higher severity of respiratory diseases. I suggest Martu et. Al. Rom J. Oral Rehab 2020, 12, 116.

R2. In the revised Introduction section, we now discuss microbial changes associated with oral dysbiosis, and also provide evidence linking oral and lung dysbiosis in specific respiratory diseases (page 2, lines 20-24; page 3, line: 1). In addition, we have provided evidence linking periodontitis and SARS-CoV-2 infection, which is of clinical relevance given the recent COVID-19 pandemic. New references have been added to the revised manuscript (Martu et al. [13] and Molina et al. [14]) accordingly.

Q3. Did the authors perform power analysis?

R3. We thank the Reviewer for bringing this up. First, our sample size was limited to the challenges of recruiting healthy young subjects to undergo invasive procedures (including a research bronchoscopy) in our study. To put this into perspective, we needed approximately two years to recruit 25 volunteers. Second, the main purpose of this study is to describe the oral and lower airway microbiota in healthy young persons (age: 19-30 years), as this data is scarce in the literature. We should also mention that, given the descriptive nature of our study, a sample size calculation and power analysis became even more complex. Moreover, the best approach to estimate a sample size calculation when performing microbiome analysis remains unclear. We have added a sentence in the Discussion section emphasizing this limitation (page 15, lines: 1-5).

Q4. Data in the results section could be better systematized. An important part of the data is repeated in the discussions section. The authors should try to be as synthetic and clear as possible in the results section.

R4. The Results section, including its tables and figures, has been completely rearranged, making it easier for the reader to understand the data presented in our manuscript.

Q5. The discussions section is lacking in comparison with the existing literature. The authors should include more studies on the subject.

R5. We thank the reviewer for pointing this out. In the revised text, we have included a section (1st paragraph of Discussion) wherein we summarize the data from Dickson et al. [34] and Ramsheh et al. [35] studies, which both contained data on “healthy” control subjects.

Round 2

Reviewer 1 Report

Manuscript well-improved.

Reviewer 3 Report

The manuscript has been improved, however, the references still must be written in the MDPI style.